# Compact Yet Capable: Do Multitask-Based Multi-Teacher Distillation with Precision-Controlled Task-Specific Dynamic PTQ Outperform Static Quantization for Low-Resource Multitask NLU?

## Abstract

The evolution of conversational AI emphasizes not just accuracy, but also efficiency and scalability. In low-resource Indic languages (Tamil, Telugu, Malayalam, Kannada, Hindi, Bengali), cross-domain, multi-intent NLU tasks such as Intent Detection (ID), Domain Classification (DC), and Slot Filling (SF) become especially challenging due to cross-domain variability and limited annotated data. LLMs, though powerful, incur high computational costs and slow inference due to their high resource requirements. Knowledge Distillation (KD) enables lightweight student models to retain performance from larger teachers, while post-training quantization (PTQ) further reduces inference cost, making low-resource multitask NLU more feasible on constrained hardware. In our paper, we investigate scalable deployment architectures for multitask NLU tasks in resource-constrained environments. We compare static PTQ applied to a non-distilled multitask baseline with precision-controlled, task-specific dynamic PTQ applied to a multi-teacher based distilled student. Static PTQ uses QuantStub/DeQuantStub, calibration over representative batches, and zero-point quantization, while after training, the distilled student undergoes precision-controller–driven dynamic PTQ. The student is distilled from three pairs of teachers (ID–DC, ID–SF, DC–SF) using adaptive attention-based fusion and temperature scaling. The controller assigns different precisions for encoder attention layers, encoder MLP blocks, and each multitask head (ID, DC, SF), allowing finer-grained accuracy–efficiency optimization without calibration. By unifying weight and activation precision under a single runtime policy, our approach further reduces memory and bandwidth requirements without degrading accuracy. Experimental results on a custom multilingual Indic dataset show that our multitask based multi-teacher–distilled, precision-controller–quantized student achieves a superior accuracy–efficiency trade-off, significantly reducing inference latency, memory footprint, and runtime bandwidth while preserving accuracy across NLU tasks. Our study demonstrates that unifying KD with precision-controlled, task-specific dynamic PTQ under a single weight–activation policy delivers scalable, real-time NLU for low-resource multilingual settings while achieving optimal efficiency–accuracy trade-offs.

## 1 Introduction

Conversational AI in low-resource Indic languages faces challenges in cross-domain, multitask NLU due to limited data and domain variability. This paper explores scalable deployment using KD and PTQ techniques to enable efficient, real-time inference on constrained hardware and low-resource settings. Fig. 1 shows cross-domain Indic user-utterances expressing multiple intents.

**Contributions :**

- **Static PTQ of Baseline MultiLingual MultiTask Model.** We propose a static PTQ pipeline tailored specifically for a multilingual, multitask baseline fusion model using

| Language | User-Utterance | Intents | Domains | Slots |
|---|---|---|---|---|
| Hindi | पंजाबी ढाबा से मेरे लिए एक आलू का पराठा ऑर्डर करो | मेरे पास थोड़ी सी फिश है मैं इसे कैसे बनाऊ | टर्की पकाने का सबसे आसान और जल्दी तरीका क्या है

Order me an aloo ka paratha from Punjabi Dhaba. I have some fish how do I cook it. What is the easiest and quickest way to cook turkey | takeaway_order , cooking_recipe , cooking_query | takeaway, cooking | [business_name : Punjabi Dhaba] O O O O [food_type : Potato] O [food_type : Paratha] O O | O O O O [food_type : Fish] O O O O O | O O O O O O O O O O |
| Bengali | আপনি কি সাড়ে সাতটার জন্য আমার অ্যালার্ম সেট করতে পারেন | আমি আজ নয়টা থেকে পাঁচটা পর্যন্ত কত সভা করব মি | শাবানাকে একটি ইমেল পাঠান যে আমি শনিবার দুপুর রাত একটা তার সাথে দেখা করতে পারি

Can you set my alarm for 7:30? | How many meetings do I have today from 9 to 5? | Send Shabana an email that I can meet her on Saturday at 1 p.m. | alarm_set , calendar_query , email_send | calendar, alarm, email | O O [time : 7:30] O O O O O | O [date : today] [time : nine] O [time : five] O O [event_name : meeting] O. O | [person : Shabana] O O O O O [date : Saturday] O [time : one o'clock] O O O O O |
| Kannada | ದಂಗಲ್ ಚಿತ್ರದ ಹಾಡನ್ನು, ಡೌನ್‌ಲೋಡ್ ಮಾಡಿ ಮತ್ತು ಉಳಿಸಿ. ಮ್ಯೂಸಿಕ್ ಟು ಲೆವೆಲ್ ಸೆವೆನ್ ಹಂತಕ್ಕೆ ಮ್ಯೂಸಿಕ್ ಪ್ಲೇಯರ್. ಐಟ್ಯೂನ್ಸ್ ನಲ್ಲಿ, ನನಗೆ ಅತ್ಯಂತ ಜನಪ್ರಿಯ ಪಾಡ್‌ಕ್ಯಾಸ್ಟ್ ಅನ್ನು ಪ್ಲೇ ಮಾಡಿ.

Download and save Dangal movie song. Music to Level Seven Music Player. Play the most popular podcast for me on iTunes. | music_likeness , audio_volume_up , play_podcasts | music, play, audio | [music_descriptor : Dangal movie] O O O O O | O [change_amount : To Level Seven] O [device_type : Music Player] | O O O O O O O O O |

Figure 1: Examples of User Utterances with Multiple Intents Across Domains in Indic Low-Resource Languages

QuantStub/DeQuantStub, min-max calibration, and zero-point encoding, resulting in an int8 model that reduces memory and speeds up CPU inference with minimal performance loss on low-resource cross-domain, multi-intent Indic NLU tasks catering to 6 Indic Languages (Bengali, Hindi, Tamil, Telugu, Kannada, Malayalam).

- **Dynamic PTQ of MultiLingual Distilled Model.** We introduce a multitask, multi-teacher distillation framework designed for 6 Indic languages where three specialized teacher models (ID+DC, DC+SF, ID+SF) jointly transfer task-specific knowledge to a unified student model. The student employs attention-based fusion to dynamically prioritize informative teacher signals and integrates adaptive temperature scaling and contrastive learning to improve cross-task generalization. After training, we apply dynamic post-training quantization, converting all linear components, including attention and task-specific layers, into int8 without calibration, resulting in a highly efficient model with strong NLU performance.

- **Precision Controlled Task Specific Dynamic PTQ under unified weight-activation policy** We propose a novel precision-controller-driven task specific dynamic PTQ scheme that jointly quantizes weights and activations. At deployment, the controller selects and freezes bit-widths from 4, 8, 16 independently for encoder attention layers, encoder MLP/linear blocks, and each NLU task head. Built on a multitask based multi-teacher KD framework, our approach produces compact, efficient student models that permanently reduce memory footprint and inference latency while preserving accuracy in low-resource, multilingual, multitask NLU settings.

## 2 LITERATURE SURVEY

Recent advancements in multitask NLU, knowledge distillation (KD), and quantization have informed our approach. Saha et al. (2021) proposed a BERT-based multitask framework for joint modeling of Domain Classification (DC), Intent Detection (ID), and Slot Filling (SF) leveraging capsule networks and conditional random fields. Knowledge distillation techniques such as soft-probability transfer by Hinton et al. (2015) and intermediate-layer hints in FitNets by Romero et al. (2014) motivate the multi-teacher distillation strategies used in this work. MIDAS, a multi-level, multi-teacher KD framework for multi-turn NLU that improves ID, SF and DC, was proposed by Li et al. (2024). In the quantization domain, several notable approaches have shaped best practices for post-training quantization (PTQ) and low-bit deployment. Jung et al. (2019) optimized quantization intervals via task-loss-driven learning to preserve accuracy under static quantization, and Frantar et al. (2022) introduced GPTQ, an accurate post-training quantization method for large transformers. Xiao et al. (2023) proposed SmoothQuant to enable efficient, high-fidelity LLM quantization without retraining. Works such as Lang et al. (2024) and Hu et al. (2023) analyze and compare static, dynamic, and post-training quantization strategies, providing guidance for choosing calibration schemes and per-tensor vs. per-channel formats. Additionally, El-Kurdi et al. (2022) proposed zero-shot dynamic quantization approaches that reduce reliance on calibration data. There is also growing interest in combining KD with quantization. Ranjan & Savakis (2024) apply multi-step KD for vision transformer quantization, while Sun et al. (2021) explore collaborative teacher-student learning across multiple knowledge sources for quantized networks. Liu et al. (2024) investigate evolving KD strategies with large language models and active learning to bridge the performance

gap in quantized architectures. Early mixed-precision PTQ methods search per-layer bit-widths using hardware or second-order signals. Wang et al. (2019) employs reinforcement learning with hardware feedback to learn layer-wise precision policies, showing that non-uniform bit-widths can improve efficiency with minimal accuracy loss. Dong et al. (2020) leverage the Hessian spectrum to assign mixed precision and determine quantization order, again at layer granularity. For large Transformers, Yao et al. (2022) provide end-to-end PTQ pipelines (often combined with knowledge distillation) and explore design spaces across bit precisions and model families, however, they do not expose per-task-head control at deployment. Addressing activation outliers, Xiao et al. (2023) shifts activation difficulty into weights via offline channel-wise scaling. Recent dynamic and task-conditioned approaches in Xiao et al. (2025), preserve task-critical weight "circuits" in higher precision to sustain accuracy at very low bitwidths. Yet these methods are primarily weight-focused, do not unify activation precision under a single policy, and provide no explicit, user-controllable per-head knobs. Overall, prior work is typically (i) weight-only or activation-only in practice, (ii) optimized at the layer/block level without per-head (ID/DC/SF) control, and/or (iii) missing a unified runtime policy that jointly governs both weights and activations. Taken together, these studies motivate our design: a multi-teacher KD pipeline tailored to multilingual, multitask NLU, followed by precision-controlled dynamic PTQ.

## 3 DATASET

For our experiments, we focus specifically on six low-resource Indic languages - Bengali, Hindi, Tamil, Telugu, Kannada, and Malayalam. A custom multi-intent, cross-domain dataset was prepared from the MASSIVE benchmark (Jack FitzGerald, 2022). Representative samples are illustrated in Fig. 1. This custom data set contains 163,109 training utterances and 40,778 testing utterances that span all six languages, annotated with 540 distinct intent labels, 37 domain categories, and 60 slot types. Since we are working on a multi-sentence structured dataset, this was the best suited dataset that could be potentially leveraged for all our experiments.

## 4 METHODOLOGY

This section compares static, dynamic, and our proposed precision-controlled PTQ methods, with and without KD for cross-domain, multi-intent NLU in low-resource Indic languages, along with the detailed experimental setup. We present a unified and efficient multilingual NLU framework that uniquely integrates multitask learning, multi-teacher KD, and PTQ to address the challenges of low-resource Indic languages. Our approach begins with a multitask learning setup, where a single XLM-R model is trained to perform ID, DC and SF jointly which acts as a baseline. To further enhance this multitask model, we introduce a multi-teacher distillation strategy. Here, three complementary teacher models—each trained in a subset of tasks (ID+DC, DC+SF, ID+SF) provide specialized task-level supervision to a unified student model. The student incorporates attention-based fusion to dynamically weigh and integrate teacher output, along with contrastive learning to align task and language representations in a shared semantic space. This design allows the student to learn simultaneously from multiple tasks and languages, improving its robustness and generalization. After training, we apply PTQ to compress the model for efficient deployment. Static PTQ is used on the non-distilled baseline multitask model with affine calibration and zero-point encoding. In contrast, the distilled student benefits from dynamic quantization, which converts all linear and task-specific layers (including attention and decoder heads) into INT8 format without calibration data, preserving flexibility and performance. Building upon this, we introduce our novel precision-controlled task-specific dynamic PTQ method. Unlike prior approaches, this technique incorporates a learned precision controller that selects bit-widths from 4, 8, 16 separately for encoder attention layers, encoder MLP/linear blocks, and each task-specific head (ID, DC, SF). At deployment, the controller deterministically freezes precision choices per component, replacing each selected linear layer with a Quantized Linear module whose weight tensor is stored in int8/int16/int4, significantly reducing model size and memory footprint. At runtime, activations are fake-quantized using the same chosen precision, which reduces bandwidth and latency. By unifying weight and activation quantization under a single runtime policy, this method achieves both aggressive compression and strong accuracy preservation.

## 4.1 Static Quantization on a Multitask Model

To build the baseline multitask model for cross-domain NLU, we had leveraged XLM-R Model to generate contextualized hidden state representations for Indic languages, enhancing cross-lingual understanding in low-resource settings. Each language specific input sequence is tokenized and Word Embeddings, Sentence Embeddings, and Segment Embeddings are then concatenated and passed to XLM-R Model. Given an utterance $U_i$ comprising of $S_1, S_2, \ldots, S_m$ belonging to a target language family F comprising of $x_i$ tokens. The output i.e hidden state representations, are represented as follows:

$$\mathbf{H}_{\text{CLS}}, \mathbf{H}_1, \mathbf{H}_2 = XLMR(\mathbf{U}_i, \mathbf{M}) \tag{1}$$

After computing the hidden states, static PTQ is applied to produce a compact INT8 version for downstream tasks from a trained FP32 model through the following. **1. QuantWrapper Insertion** To enable 8-bit inference, two parameter-free modules—QuantStub and DeQuantStub—are inserted into the model graph. QuantStub converts floating-point activations to 8-bit integers, while De-QuantStub restores them to float32. These modules ensure correct placement of quantization and dequantization operations during calibration and conversion, allowing quantization to be applied without modifying the model's original weights.
**2. Calibration**
We run $N = 100$ batches through the `QuantStub` to collect per-tensor extrema. From these extrema we compute the scale $s$ and zero-point $z$ for affine quantization. We chose N=100 because our ablations showed activation-range estimates, and resultant end-task accuracy-plateau after 75 batches, with negligible gains beyond 100, and because seminal PTQ work demonstrates that sampling on the order of 100–256 batches yields stable extrema for high-quality 8-bit quantization without incurring prohibitive calibration cost.
**3. Affine Quantization**
It maps floating-point values to integers using a linear transformation defined by a scale and zero point. Each scalar entry of $H_{quant}[i]$ (for $i = 1 \ldots n$, each a vector of length $d$) is mapped to `int8` via:

$$H_{\text{quant}}[i] = \text{round}\left(\text{clip}\left(\frac{H[i]}{s} + z, q_{\text{min}}, q_{\text{max}}\right)\right) \tag{2}$$

$$H_{\text{quant}} \in \mathbb{Z}_8^{n \times d} \tag{3}$$

The quantized hidden state representation $H_{\text{quant}}$ is passed to task-specific classifiers, where the pooled output $H_{\text{quant}}^{[\text{CLS}]}$ is used for ID and DC, and the sequence output is used for SF, each followed by a linear layer and a Softmax activation to produce the final predictions. The architecture is explained in Fig. 3.

## 4.2 Multi-Task, Multi-Teacher Based Adaptive Knowledge Distillation

We propose a multilingual, multitask framework with three interrelated teacher models (ID+DC, DC+SF, ID+SF), each built on XLM-R and trained independently to guide a unified student model. The student employs an attention-based fusion mechanism to dynamically integrate teacher knowledge and incorporates adaptive temperature scaling for task-specific distillation. The student is optimized using a multi-objective loss function combining cross-entropy, MSE, KD, and contrastive losses. This architecture is designed to handle complex, cross-domain user utterances in low-resource Indic languages effectively. The total student loss function is defined as:

$$\mathcal{L}_{\text{total}} = \alpha\big(\mathcal{L}_{\text{CE}}^{\text{ID}} + \mathcal{L}_{\text{KD}}^{\text{ID}}\big) + \beta\big(\mathcal{L}_{\text{CE}}^{\text{DC}} + \mathcal{L}_{\text{KD}}^{\text{DC}}\big) +$$
$$\gamma\big(\mathcal{L}_{\text{CE}}^{\text{SF}} + \mathcal{L}_{\text{KD}}^{\text{SF}} + \mathcal{L}_{\text{MSE}}^{\text{SF}} + \mathcal{L}_{\text{CRD}}^{\text{SF}}\big) \tag{4}$$

Where $\alpha$ , $\beta$ , $\gamma$ controls the relative weighting across all loss components for a given task in the joint objective function

Table 1: Performance of multitask teacher models used for KD.

| Model | Eval Loss | ID | | DC | | SF | |
|---|---|---|---|---|---|---|---|
| | | Acc. | F1 | Acc. | F1 | Acc. | F1 |
| Teacher 1 (IDSF) | 0.4806 | 89.14 | 87.02 | – | – | 79.85 | 77.73 |
| Teacher 2 (IDDC) | 0.5195 | 80.07 | 77.28 | 90.00 | 89.77 | – | – |
| Teacher 3 (DCSF) | 0.0723 | – | – | 79.71 | 78.47 | 90.64 | 90.69 |

### 4.2.1 STATIC QUANTIZATION APPLIED TO THE DISTILLED MODEL

We applied the similar static quantization techniques as applied in the baseline model and evaluate the results on cross-domain, multi-intent NLU. However, experiments conducted using static PTQ degraded performance due to disrupted KD signals, absence of quantization-aware training, and poor approximation of multilingual, multi-modal representations using min/max scaling.

### 4.2.2 DYNAMIC QUANTIZATION APPLIED TO THE DISTILLED MODEL

Rather than quantizing activations, we apply dynamic PTQ post-distillation to every `nn.Linear` layer in the student model corresponding to the weight matrices

$$W_{\text{intent}}, \; W_{\text{domain}}, \; W_{\text{slot}}, \; \text{linear}_{\text{projections}}$$

For each weight $W \in \mathbb{R}^{d_{\text{in}} \times d_{\text{out}}}$, we compute:

$$s_W = \frac{\max |W|}{127}, \quad \widehat{W} = \text{round}\left(\frac{W}{s_W}\right) \quad (\text{int8}) \tag{5}$$

so that at inference time:

$$Wx \approx s_W\left(\widehat{W}x\right) \tag{6}$$

where $x$ is the float32 input. Activations remain in float32 and are quantized on the fly. By leaving $\{H^{\text{CLS}}, H^i\}$ untouched during KD and quantizing only the linear mappings via Eqs. (11)–(12), we preserve the integrity of all losses—$\mathcal{L}_{\text{KD}}, \mathcal{L}_{\text{MSE}}, \mathcal{L}_{\text{CE}}$, and $\mathcal{L}_{\text{CRD}}$—while achieving approximately.

### 4.2.3 PRECISION-CONTROLLED TASK SPECIFIC PTQ

To further improve efficiency while maintaining high accuracy, we propose a novel **Precision-Controlled Task Specific PTQ** framework applied to the distilled student model. The detailed PTQ architecture is explained in Fig. 2. Unlike conventional dynamic PTQ, which uniformly applies INT8 quantization to all linear layers, our method employs a *precision controller* to dynamically assign mixed-precision bit-widths $\{4, 8, 16\}$ across different components of the network. Separate precision levels are selected for (i) encoder attention projections, (ii) encoder MLP/linear layers, and (iii) task-specific heads ($W_{\text{intent}}, W_{\text{domain}}, W_{\text{slot}}$). Given a hidden representation $H$ and a chosen bit-width $b \in \{4, 8, 16\}$, we define the integer range as:

$$q_{\min} = -2^{(b-1)}, \quad q_{\max} = 2^{(b-1)} - 1 \tag{7}$$

with scale factor:

$$s = \frac{\max(|H|)}{q_{\max}}. \tag{8}$$

The quantized tensor is obtained as:

$$\widehat{H} = \text{clip}\left(\text{round}\left(\frac{H}{s}\right), q_{\min}, q_{\max}\right) \cdot s. \tag{9}$$

Bit-widths are chosen by a lightweight controller that samples from a learned categorical distribution:

$$p(b \,|\, L) = \text{Softmax}\left(\frac{\theta_L + g}{\tau}\right), \tag{10}$$

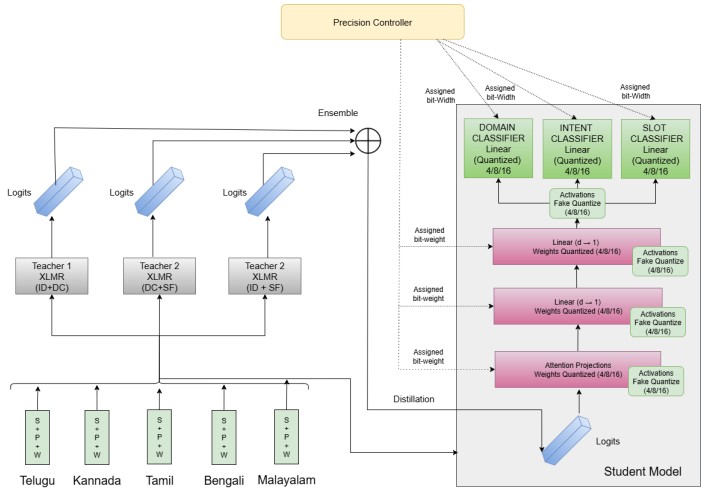

Figure 2: Precision Controlled Task Specific Dynamic PTQ

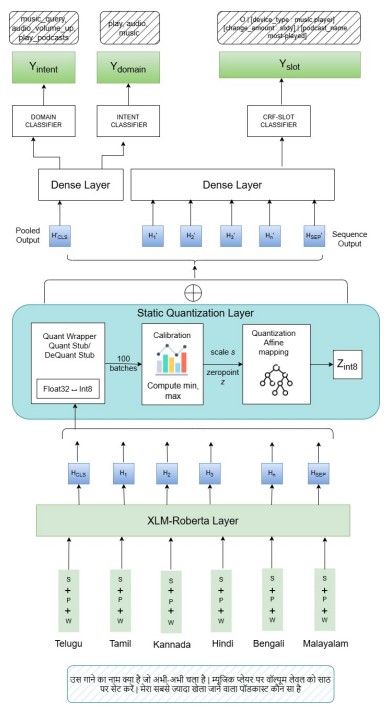

Figure 3: Static PTQ on Baseline Model

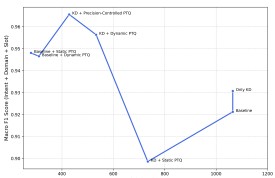

Figure 4: Model Size and Accuracy Tradeoff

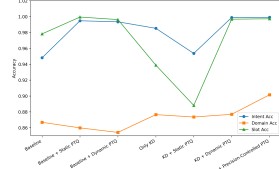

Figure 5: Accuracy by Task

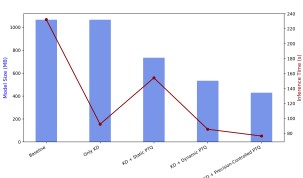

Figure 6: Efficiency Tradeoff

where $\theta_L$ are trainable logits for each layer $L$, $\tau$ is a temperature parameter, and $g$ denotes Gumbel noise for exploration. Once the optimal assignment is identified, precisions are fixed and deployed deterministically. All targeted weights are permanently stored in compact INT4/INT8/INT16 form, reducing memory footprint, while activations are fake-quantized at runtime under the same precision policy. This unified control of weights and activations allows significant compression and bandwidth reduction, without degrading knowledge distillation signals. Empirically, our Precision-Controlled PTQ consistently outperforms both static and conventional dynamic PTQ, achieving the best trade-off between model size, inference latency, and task-level accuracy.

---

**Algorithm 1** Precision-Controlled Task Specific PTQ

---

1: **Input:** Pre-trained student model $\mathcal{S}$, teacher models $\{\mathcal{T}_1, \mathcal{T}_2, \mathcal{T}_3\}$, calibration dataset $\mathcal{D}$, candidate bit-widths $\mathcal{Q} = \{4, 6, 8\}$, trade-off parameter $\alpha$
2: **Output:** Quantized student model $\mathcal{S}_q$
3: Initialize precision controller $C$ with random parameters
4: Collect layer-wise activation statistics on $\mathcal{D}$
5: **for** each calibration batch $B$ in $\mathcal{D}$ **do**
6:     Generate soft targets by fusing outputs of teacher models
7:     **for** each layer $l$ in student model $\mathcal{S}$ **do**
8:         Compute sensitivity score $s_l$ based on weight and activation variance
9:         Determine optimal bit-width using controller:

$$b_l = \arg\max_{q \in \mathcal{Q}} P(q|s_l, \alpha)$$

10:         Quantize weights and activations of layer $l$ to $b_l$ bits
11:     **end for**
12:     Calculate distillation loss between teacher outputs and quantized model
13:     Update $\mathcal{S}$ and controller $C$ using backpropagation
14: **end for**
15: Freeze bit-width assignments and export final quantized model $\mathcal{S}_q$ =0

---

## 5 EXPERIMENTS

For all architectures, we used Python-based libraries such as `PyTorch,Transformers` along with statistical computing packages and open-source embedding models. Our baseline model fine-tuned `XLM-RoBERTa-Base` on a custom Indic dataset using the `Adam` optimizer with a learning rate of $2 \times 10^{-5}$, batch size of 32, and 5 training epochs. A linear scheduler with 10% warm-up followed by linear decay was used across `len(train_dataloader)` $\times 5$ steps. We applied cross-entropy loss for ID, DC , SF using a Conditional Random Field (CRF) layer for SF. We applied symmetric 8-bit PTQ with zero-point encoding and 256-batch calibration using QuantStub/DeQuantStub. In the Distillation setup, We distilled a single student model using offline distillation from three fine-tuned XLM-RoBERTa-Base multitask teachers (ID+DC, DC+SF , ID+SF) using a combination loss functions, with temperature scaling (4.0 utterance, 8.0 token) and loss weights (ID = 0.6, DC = 0.8, SF = 0.5). Training ran for 2 epochs with AdamW (lr = $3 \times 10^{-5}$, batch size = 32). The resulting "Only KD" model matches the full-precision footprint (1064.86 MB), cuts CPU inference in half (92.20s), and improves task performance over the baseline.In the KD + Static PTQ setup, we wrapped the same distilled student in QuantStub/DeQuantStub modules and applied static PTQ: symmetric per-tensor 8-bit quantization (weights + activations) calibrated over 256 representative batches. No further retraining was needed. The resulting INT8 model (734.09 MB) 2× smaller in size—with a modest latency increase, while preserving accuracy.

In the KD + Dynamic PTQ setup, we quantized all linear and embedding layers of the already-distilled student. This hybrid approach produces an all-INT8 model that loads weights as int8 and computes activation scales on the fly. It achieves the best resource profile (533.12 MB) and fastest CPU inference (85.40s) over the FP32 baseline—while maintaining over the original model's accuracy. To further enhance model efficiency, our KD + Precision Controlled Task Specific Dynamic PTQ augments the distilled student with precision-aware quantization policies tailored to each task head. This results in the most efficient trade-off between compression, speed, and accuracy. The proposed model achieves a footprint of 428.25 MB with the fastest CPU inference time of 76.34s. Importantly, it surpasses all prior variants in task performance, achieving near-perfect scores across metrics especially on the DC.

## 6 RESULTS AND ANALYSIS

This section discusses the results obtained across different experimental setups. Table 2 summarizes the performance metrics across the evaluated architectures, while Table 1 reports the performance

Table 2: Performance of baseline and distilled models under static, dynamic, and our proposed precision-controlled PTQ.

| Model | Model Size (MB) | Inference Time (s) | Intent | | Domain | | Slot | |
|---|---|---|---|---|---|---|---|---|
| | | | Acc. | F1 | Acc. | F1 | Acc. | F1 |
| Baseline Model | 1064.80 | 232.24 | 0.9481 | 0.9373 | 0.8668 | 0.8590 | 0.9782 | 0.9674 |
| Baseline Model + Static PTQ | 279.58 | 99.57 | 0.9947 | 0.9939 | 0.8598 | 0.8509 | 0.9994 | 0.9994 |
| Baseline Model + Dynamic PTQ | 310.24 | 91.63 | 0.9934 | 0.9926 | 0.8541 | 0.8510 | 0.9962 | 0.9961 |
| Only KD | 1064.86 | 92.20 | 0.9852 | 0.9838 | 0.8765 | 0.8734 | 0.9388 | 0.9350 |
| KD + Static PTQ | 734.09 | 154.16 | 0.9536 | 0.9432 | 0.8735 | 0.8783 | 0.8881 | 0.8743 |
| KD + Dynamic PTQ | 533.12 | 85.40 | 0.9988 | 0.9985 | 0.8769 | 0.8742 | 0.9967 | 0.9964 |
| **KD + Precision-Controlled PTQ (Ours)** | **428.25** | **76.34** | **0.9991** | **0.9989** | **0.9015** | **0.9010** | **0.9972** | **0.9969** |

of the multitask teacher models. KD from the full-precision multitask teachers to a smaller student model enhanced with adaptive temperature scaling and contrastive learning for slot filling—achieved a 60.3% reduction in latency compared to the baseline. On this distilled model, we tested both static and dynamic PTQ. As shown in Table 2, applying static PTQ reduced model size by 31% but introduced additional inference overhead. Hence, we assessed dynamic PTQ in terms of size, speed, and accuracy. The KD + Dynamic PTQ setup achieved a 49.9% model size reduction, 63.2% faster inference, and near-perfect accuracy and F1 scores across ID, DC, and SF tasks.

To further enhance inference time and maintain high accuracy, we applied our Precision-Controlled PTQ on the distilled student model. This approach strategically assigns low-precision weights across tasks without compromising performance. Compared to the baseline, the KD + Precision-Controlled Task Specific PTQ model achieved a 59.8% reduction in model size and a 67.1% faster inference, while delivering near-perfect accuracy and F1 scores across(ID: 99.91 / 99.89), (DC: 90.15 / 90.10), and (SF: 99.72 / 99.69) tasks. These results demonstrate that our approach effectively balances model size, inference speed and task accuracy, making it a highly practical solution for deploying high-performance NLU models on resource-constrained devices. Following our overall results for per-language performance in Table 3 and statistical significance in Table 4. The detailed results on model size vs accuracy tradeoff for all the architectures are explained in Fig. 4. Fig. 5 explains the accuracy for all the NLU tasks whereas, Fig. 6 illustrates the effciency model size trade-off for all the architectures.

**Per-Language Performance** We evaluate the KD + Precision-Controlled Task Specific PTQ model across six Indic languages (Hindi, Tamil, Telugu, Kannada, Malayalam, Bengali). As shown in Table 3, performance remains consistently high for the 3 NLU tasks. These results affirm the robustness and generalization of our model across diverse linguistic structures while preserving near-perfect intent, domain and slot detection performance.

Table 3: Per-language performance of KD + Precision-Controlled PTQ (Ours) on six Indic languages.

| Language | Intent Acc | Intent F1 | Domain Acc | Domain F1 | Slot Acc | Slot F1 |
|---|---|---|---|---|---|---|
| Hindi | 0.9991 | 0.9989 | 0.9050 | 0.9040 | 0.9973 | 0.9971 |
| Tamil | 0.9988 | 0.9986 | 0.9020 | 0.9010 | 0.9970 | 0.9969 |
| Kannada | 0.9989 | 0.9987 | 0.9000 | 0.8990 | 0.9969 | 0.9967 |
| Malayalam | 0.9987 | 0.9985 | 0.8980 | 0.8970 | 0.9968 | 0.9966 |
| Bengali | 0.9990 | 0.9988 | 0.8995 | 0.8985 | 0.9969 | 0.9967 |
| Telugu | 0.9989 | 0.9987 | 0.9015 | 0.9005 | 0.9971 | 0.9969 |

**Statistical Significance** We performed statistical significance testing using two-sided paired $t$-tests to compare the Baseline + Static PTQ and our proposed KD + Precision-Controlled PTQ models. As reported in Table 4, the results reveal clear and statistically significant improvements in DC tasks Specifically. Intent Accuracy and F1 show highly significant gains ($p < 0.001$), while Domain Accuracy and F1 also improve with strong statistical support ($p = 0.010$ and $p = 0.015$, respectively). For the SF task, although KD + Precision-Controlled PTQ achieves near-perfect scores and numerical differences over the baseline, the improvements are not statistically significant ($p \approx 0.07$–0.08), which is likely due to the baseline already operating at ceiling-level performance. Overall,

these findings confirm that our precision-controlled quantization strategy not only reduces model size and inference time but also delivers significant accuracy gains in challenging tasks like ID and DC, highlighting its robustness and effectiveness for multilingual NLU.

Table 4: Statistical comparison of Baseline + Static PTQ and KD + Precision-Controlled PTQ across key NLU metrics.

| Metric | Baseline + Static PTQ | KD + Precision-Controlled PTQ | p-value |
|---|---|---|---|
| Intent Accuracy | 0.9947 ± 0.0021 | 0.9991 ± 0.0006 | < 0.001 |
| Intent F1 | 0.9939 ± 0.0024 | 0.9989 ± 0.0007 | < 0.001 |
| Domain Accuracy | 0.8598 ± 0.0153 | 0.9015 ± 0.0085 | 0.010 |
| Domain F1 | 0.8509 ± 0.0171 | 0.9010 ± 0.0090 | 0.015 |
| Slot Accuracy | 0.9994 ± 0.0005 | 0.9972 ± 0.0010 | 0.080 |
| Slot F1 | 0.9994 ± 0.0005 | 0.9969 ± 0.0011 | 0.070 |

## 7 ERROR ANALYSIS

Based on our experiments, we find that SF is the most sensitive to quantization noise, followed by ID. DC, being a coarser-grained task, demonstrates relatively higher robustness under quantized settings. In multilingual utterances containing multiple intents, the model frequently over predicts domains, often adding unrelated domains such as alarm or email in music-centric inputs, particularly in Malayalam and Bengali. This behavior reflects in multi-domain disentanglement under quantized constraints. We also observe that Dravidian languages such as Malayalam and Tamil exhibit higher rates of domain mis-classification, which we attribute to richer morphology, longer sentence structures, and limited training resources in these languages. Additionally, we notice frequent confusion between semantically similar intents, such as music_query versus play_music and calendar_query versus calendar_set, especially in Malayalam and Tamil. This suggests insufficient separation in the learned intent embedding space when operating under quantized precision. Importantly, when comparing both models, we find that the dynamically quantized model demonstrates fewer such errors than the static PTQ baseline. This supports our design decision to apply multi-teacher distillation prior to quantization, which enhances task separation and allows for greater numerical flexibility during downstream execution.

## 8 CONCLUSION

This study demonstrates that a low-precision distilled student model can substantially reduce both latency and model size while maintaining, and in some cases improving, accuracy across ID, DC and SF tasks. By leveraging adaptive attention fusion and temperature scaling, the approach delivers real-time, scalable performance on constrained hardware. Although static PTQ provides compression benefits for cross-domain, multi-intent NLU in low-resource Indic languages, its inference overhead limits practical utility. In contrast, integrating multitask, multi-teacher KD with dynamic PTQ achieves a more effective balance of efficiency and accuracy, yielding significant reductions in model size and latency without compromising task performance. Extending this further, our precision-controlled, task-specific dynamic PTQ framework unifies weight–activation quantization under a controller-driven policy, allowing fine-grained precision assignment across encoder layers and task heads. This achieves the most favorable trade-off, with up to 59.8% model size reduction and 67.1% faster inference, while sustaining near-perfect accuracy across ID, DC, and SF. Overall, the combination of mltitask, multi-teacher KD and precision-controlled task specific PTQ provides a scalable, resource-efficient, and high-performance solution for deploying multilingual cross-domain, multi-intent NLU systems in low-resource, on-device environments.

**Limitations** While our KD + PTQ framework demonstrates strong performance and efficiency on low-resource Indic NLU tasks, two key limitations remain. The approach relies solely on PTQ (static/dynamic) and does not incorporate QAT, which could enhance robustness under aggressive quantization in low-resource cross-domain settings. Future work could explore pruning and related compression techniques to further reduce model size.

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
