# OpenReview forum: "Compact Yet Capable: Do Multitask-Based Multi-Teacher Distillation with Precision-Controlled Task-Specific Dynamic PTQ Outperform Static Quantization for Low-Resource Multitask NLU?"
_ICLR.cc/2026/Conference — ICLR 2026 Conference Withdrawn Submission_

### Official Review · Reviewer_yVeq · 2025-10-29

**Soundness:** 2
**Presentation:** 1
**Contribution:** 1
**Rating:** 2
**Confidence:** 4

**Summary:**

This paper proposes a pipeline for creating efficient NLU models for low-resource Indic languages. The proposed method combines multi-teacher knowledge distillation with a precision-controlled dynamic PTQ scheme. The authors claim this approach achieves a superior trade-off between model size, inference speed, and accuracy on multitask NLU problems (ID, DC, SF). While the goal of improving NLU efficiency in low-resource settings is highly relevant, this paper should be rejected due to (1) a significant overstatement of its contributions, as the core techniques are largely combinations of existing work; (2) a lack of novelty in the proposed "precision controller," which is not well-differentiated from prior art; (3) unconvincing and poorly analyzed experimental results that raise questions about the dataset's validity and the reported performance gains; and (4) a poorly structured and unpolished presentation.

**Strengths:**

The paper addresses the important and practical problem of deploying multitask NLU models for low-resource languages on resource-constrained devices. The proposed end-to-end pipeline, which integrates knowledge distillation with mixed-precision quantization, is a sensible and approach to this challenge.

**Weaknesses:**

The paper's central claims of contribution are fundamentally weak and appear to be incremental at best. The authors present three contributions, yet the first two, applying static PTQ to a baseline and dynamic PTQ to a distilled model, are standard practices and cannot be considered novel contributions. The third and primary claimed contribution, a "Precision Controlled Task Specific Dynamic PTQ," is conceptually similar to existing work on mixed-precision quantization (e.g., APTQ [1], SliM-LLM [2]), but the paper fails to provide any comparison or discussion of these highly relevant prior approaches. The method of training a controller to assign bit-widths lacks algorithmic innovation and is not justified with sufficient theoretical or empirical backing against state-of-the-art methods.

Furthermore, the experimental evaluation is unconvincing. The reported near-perfect scores (often >99%) across most tasks and languages suggest that the custom dataset may lack sufficient difficulty, making it an unreliable benchmark for demonstrating the superiority of the proposed method. The puzzling result where quantization *improves* performance over an already-distilled model is counter-intuitive and left entirely unexplained, casting doubt on the reliability of the experimental setup. The paper also lacks critical ablation studies and analysis that would be necessary to support its claims.

**Major Weaknesses:**

1.  **Overstated and Unoriginal Contributions:**
    *   **Contribution 1 ("Static PTQ of Baseline...")**: Applying a standard compression technique like static PTQ to a baseline model is a routine experimental step, not a scientific contribution.
    *   **Contribution 2 ("Dynamic PTQ of... Distilled Model")**: Both multi-teacher distillation and dynamic PTQ are well-established techniques. Combining them is a straightforward engineering effort, not a novel research contribution.
    *   **Contribution 3 ("Precision Controlled... Dynamic PTQ")**: The core idea of using a controller or search algorithm to find optimal mixed-precision bit-widths is not new. The paper completely ignores highly relevant and recent work such as APTQ [1] and SliM-LLM [2]. Without a direct comparison or even a discussion, the novelty and superiority of the proposed controller remain entirely unsubstantiated.

2.  **Unconvincing Experimental Results and Lack of Analysis:**
    *   **Dataset Saturation and Potential Contamination**: The reported performance metrics are consistently near-perfect (e.g., Intent F1 of 0.9989, Slot F1 of 0.9969 in Table 2). Such high scores raise serious concerns about whether the dataset has enough challenging examples to differentiate between strong models. It is possible that the observed gains are merely artifacts of a simplistic benchmark. Furthermore, given the ceiling-level performance, it is crucial to investigate potential data contamination.
    *   **Unexplained Performance Gains from Quantization**: A major red flag is that the "KD + Dynamic PTQ" model outperforms the "Only KD" model (e.g., Intent F1 0.9985 vs. 0.9838). Quantization is an information-destroying process and should theoretically lead to an accuracy drop, however small. This counter-intuitive result is not analyzed or explained, suggesting potential issues with the experimental setup, evaluation protocol, or hyperparameter tuning.
    *   **Missing Ablation Studies**: The paper lacks crucial analysis. For instance:
        *   How do different teachers in the multi-teacher setup contribute to the final performance? An ablation study on the teachers is missing.
        *   How does the proposed "precision controller" compare against simpler heuristics or even random search for bit-width assignment?
        *   How does the model generalize to other Indic languages not included in the training set?
    *   **Hyperparameter Sensitivity**: Equation (4) presents a complex loss function with multiple components and weighting hyperparameters (`α, β, γ`). The paper provides no details on how these were tuned or an analysis of their sensitivity, which is critical for reproducibility and understanding the model's behavior.

3.  **Limited Scope and Scalability Concerns:**
    *   The experiments are conducted on relatively small models (XLM-R-Base). The true challenge of quantization lies in its application to massive models (100B+ parameters). The paper provides no evidence that the proposed controller-based approach would scale effectively to such models, where the search space for precision assignment would be astronomically larger.

4.  **Poor Structure and Presentation:**
    *   **Introduction**: The introduction is sparse, lacks proper context-setting, and contains no citations to ground the work in existing literature.
    *   **Disorganized Sections**: The paper's structure is illogical. The dataset description (Section 3) should be part of the experimental setup, not a standalone section before the methodology.
    *   **Missing Citations**: The methodology section makes vague references to "prior approaches" (e.g., line 154) without citing any specific papers.
    *   **Formatting Issues**: The document is rife with formatting issues that impede readability. For example, the list of PTQ steps on line 174 is poorly formatted and squeezed together. Figures mentioned on page 3 do not appear until page 6, forcing the reader to jump back and forth.

[1]: APTQ: Attention-aware Post-Training Mixed-Precision Quantization for Large Language Models

[2]: SliM-LLM: Salience-Driven Mixed-Precision Quantization for Large Language
Models

**Questions:**

See weaknesses

---

### Official Review · Reviewer_23Nm · 2025-10-31

**Soundness:** 2
**Presentation:** 2
**Contribution:** 2
**Rating:** 2
**Confidence:** 4

**Summary:**

The paper proposes a novel framework for efficient multitask NLU in low-resource Indic languages. It combines multi-teacher knowledge distillation with a precision-controlled, task-specific dynamic PTQ method. The core innovation is a controller that assigns different bit-widths (4, 8, 16) to encoder layers and individual task heads (ID, DC, SF), unifying weight and activation quantization under one policy. Experiments show this approach significantly reduces model size and inference time while improving or matching accuracy compared to static PTQ.

**Strengths:**

1.Strong Problem Focus:Addresses the critical need for efficient, multilingual NLU on constrained hardware.
2.Innovative Quantization:The precision controller enabling fine-grained, task-specific bit-width assignment is a key contribution.

**Weaknesses:**

1.No QAT Comparison: Lacks comparison against Quantization-Aware Training, which might yield better results for sensitive tasks like Slot Filling.
2.Limited Controller Insight: Doesn't deeply explain how the controller learns its optimal bit-width assignments.
3.Hardware Scope:Evaluation is limited to CPU; performance on other edge devices is unknown.

**Questions:**

1.  How is the precision controller trained? Is it joint with the student model?
2.  What specific signals guide the controller's bit-width selection for different components?
3.  How does this method compare directly to state-of-the-art PTQ techniques like GPTQ or SmoothQuant?
4.  Would a simpler KD setup  still benefit from the proposed PTQ?

---

### Official Review · Reviewer_eWdF · 2025-11-01

**Soundness:** 1
**Presentation:** 1
**Contribution:** 1
**Rating:** 2
**Confidence:** 4

**Summary:**

To reduce the inference cost of multilingual, multi-task language models, the authors propose a model compression framework that integrates knowledge distillation and post-training quantization.
In the knowledge distillation stage, a student model is distilled from three teacher models, each trained on different subsets of the tasks.
For post-training quantization, the authors introduce a precision-controlled quantization method that employs a controller to determine the optimal quantization bit-width for each model layer.
By combining these two techniques, the proposed approach achieves a 59.8% reduction in model size and 67.1% faster inference, while maintaining performance comparable to the original model.

**Strengths:**

1. The quantization process is well explained, with mathematical formulations that effectively clarify the underlying mechanism.
2. The experimental results are clearly presented, making it easy to understand the performance improvements achieved by the proposed methods.

**Weaknesses:**

1. The background and motivation are not described in sufficient detail. In the Introduction section, the authors are encouraged to (a) clearly explain what cross-domain multitask NLU entails and why this application requires efficient inference of language models, and (b) provide concrete data on model size and hardware memory constraints to justify the necessity of model compression.
2. The core methodology could be explained more clearly. (a) In Section 4.2, while the authors mention cross-entropy, MSE, KD, and contrastive losses, the corresponding mathematical formulations are not provided. (b) In Section 4.2.3, the introduction of trainable logits $\theta_L$ lacks a clear description of the training objective. (c) In line 8 of Algorithm 1, the sensitivity score $s$ mentioned is not defined in the text. (d) The definition of the bit-width controller differs between Equation (10) and line 9 of Algorithm 1. These inconsistencies may make it difficult for readers to fully understand the proposed approach.
3. The overall presentation of the paper could be improved for better readability. (a) In Section 2, the discussions on quantization and knowledge distillation are not clearly separated. (b) Sections 4.1, 4.2.1, and 4.2.2 include lengthy descriptions of baseline implementations. (c) In Sections 5 and 6, the experimental setup and results are not clearly separated, making the evaluation process harder to follow.
4. Some experimental results would benefit from deeper analysis. For example, in Table 2, it is unclear why applying PTQ leads to better performance compared to not applying it. Similarly, it is unclear why the inference time of static quantization being longer than that of dynamic quantization, as dynamic quantization typically requires additional computation for online parameter estimation. Providing explanations for these findings would strengthen the paper’s empirical analysis.

**Questions:**

See weaknesses.

---

### Note · Authors · 2025-11-20

I have read and agree with the venue's withdrawal policy on behalf of myself and my co-authors.